# Reinforcement Learning for Solving the Vehicle Routing Problem

**Mohammadreza Nazari**    **Afshin Oroojlooy**    **Martin Takáč**    **Lawrence V. Snyder**
Department of Industrial and Systems Engineering
Lehigh University, Bethlehem, PA 18015
`{mon314,afo214,takac,lvs2}@lehigh.edu`

## Abstract

We present an end-to-end framework for solving the Vehicle Routing Problem (VRP) using reinforcement learning. In this approach, we train a single policy model that finds near-optimal solutions for a broad range of problem instances of similar size, only by observing the reward signals and following feasibility rules. We consider a parameterized stochastic policy, and by applying a policy gradient algorithm to optimize its parameters, the trained model produces the solution as a sequence of consecutive actions in real time, without the need to re-train for every new problem instance. On capacitated VRP, our approach outperforms classical heuristics and Google's OR-Tools on medium-sized instances in solution quality with comparable computation time (after training). We demonstrate how our approach can handle problems with split delivery and explore the effect of such deliveries on the solution quality. Our proposed framework can be applied to other variants of the VRP such as the stochastic VRP, and has the potential to be applied more generally to combinatorial optimization problems.

## 1 Introduction

The *Vehicle Routing Problem* (VRP) is a combinatorial optimization problem that has been studied in applied mathematics and computer science for decades. VRP is known to be a computationally difficult problem for which many exact and heuristic algorithms have been proposed, but providing fast and reliable solutions is still a challenging task. In the simplest form of the VRP, a single capacitated vehicle is responsible for delivering items to multiple customer nodes; the vehicle must return to the depot to pick up additional items when it runs out. The objective is to optimize a set of routes, all beginning and ending at a given node, called the *depot*, in order to attain the maximum possible reward, which is often the negative of the total vehicle distance or average service time. This problem is computationally difficult to solve to optimality, even with only a few hundred customer nodes [12], and is classified as an NP-hard problem. For an overview of the VRP, see, for example, [15, 23, 24, 33].

The prospect of new algorithm discovery, without any hand-engineered reasoning, makes neural networks and reinforcement learning a compelling choice that has the potential to be an important milestone on the path toward solving these problems. In this work, we develop a framework with the capability of solving a wide variety of combinatorial optimization problems using *Reinforcement Learning* (RL) and show how it can be applied to solve the VRP. For this purpose, we consider the Markov Decision Process (MDP) formulation of the problem, in which the optimal solution can be viewed as a sequence of decisions. This allows us to use RL to produce near-optimal solutions by increasing the probability of decoding "desirable" sequences.

A naive approach would be to train an instance-specific policy by considering every instance separately. In this approach, an RL algorithm needs to take many samples, maybe millions of them, from the

underlying MDP of the problem to be able to produce a good-quality solution. Obviously, this approach is not practical since the RL method should be comparable to existing algorithms not only in terms of the solution quality but also in terms of runtime. For example, for all of the problems studied in this paper, we wish to have a method that can produce near-optimal solutions in less than a second. Moreover, the policy learned by this naive approach would not apply to instances other than the one that was used in the training; after a small perturbation of the problem setting, e.g., changing the location or demand of a customer, we would need to rebuild the policy from scratch.

Therefore, rather than focusing on training a separate policy for every problem instance, we propose a structure that performs well on any problem sampled from a given distribution. This means that if we generate a new VRP instance with the same number of nodes and vehicle capacity, and the same location and demand distributions as the ones that we used during training, then the trained policy will work well, and we can solve the problem right away, without retraining for every new instance. As long as we approximate the generating distribution of the problem, the framework can be applied. One can view the trained policy as a black-box heuristic (or a meta-algorithm) which generates a high-quality solution in a reasonable amount of time.

This study is motivated by the recent work by Bello et al. [4]. We have generalized their framework to include a wider range of combinatorial optimization problems such as the VRP. Bello et al. [4] propose the use of a Pointer Network [34] to decode the solution. One major issue that complicates the direct use of their approach for the VRP is that it assumes the system is static over time. In contrast, in the VRP, the demands change over time in the sense that once a node has been visited its demand becomes, effectively, zero. To overcome this, we propose an alternate approach—which is simpler than the Pointer Network approach—that can efficiently handle both the static and dynamic elements of the system. Our policy model consists of a recurrent neural network (RNN) decoder coupled with an attention mechanism. At each time step, the embeddings of the static elements are the input to the RNN decoder, and the output of the RNN and the dynamic element embeddings are fed into an attention mechanism, which forms a distribution over the feasible destinations that can be chosen at the next decision point.

The proposed framework is appealing to practitioners since we utilize a self-driven learning procedure that only requires the reward calculation based on the generated outputs; as long as we can observe the reward and verify the feasibility of a generated sequence, we can learn the desired meta-algorithm. For instance, if one does not know how to solve the VRP but can compute the cost of a given solution, then one can provide the signal required for solving the problem using our method. Unlike most classical heuristic methods, it is robust to problem changes, e.g., when a customer changes its demand value or relocates to a different position, it can automatically adapt the solution. Using classical heuristics for VRP, the entire distance matrix must be recalculated and the system must be re-optimized from scratch, which is often impractical, especially if the problem size is large. In contrast, our proposed framework does not require an explicit distance matrix, and only one feed-forward pass of the network will update the routes based on the new data.

Our numerical experiments indicate that our framework performs significantly better than well-known classical heuristics designed for the VRP, and that it is robust in the sense that its worst results are still relatively close to optimal. Comparing our method with the OR-Tools VRP engine [16], which is one of the best open-source VRP solvers, we observe a noticeable improvement; in VRP instances with 50 and 100 customers, our method provides shorter tours in roughly $61\%$ of the instances. Another interesting observation that we make in this study is that by allowing multiple vehicles to supply the demand of a single node, our RL-based framework finds policies that outperform the solutions that require single deliveries. We obtain this appealing property, known as the split delivery, without any hand engineering and at no extra cost.

## 2   Background

Before presenting the problem formalization, we briefly review the required notation and relation to existing work.

**Sequence-to-Sequence Models**   *Sequence-to-Sequence* models [32, 34, 25] are useful in tasks for which a mapping from one sequence to another is required. They have been extensively studied in the field of neural machine translation over the past several years, and there are numerous variants

of these models. The general architecture, which is shared by many of these models, consists of two RNN networks called the encoder and decoder. An encoder network reads through the input sequence and stores the knowledge in a fixed-size vector representation (or a sequence of vectors); then, a decoder converts the encoded information back to an output sequence.

In the vanilla Sequence-to-Sequence architecture [32], the source sequence appears only once in the encoder and the entire output sequence is generated based on one vector (i.e., the last hidden state of the encoder RNN). Other extensions, for example Bahdanau et al. [3], illustrate that the source information can be used more explicitly to increase the amount of information during the decoding steps. In addition to the encoder and decoder networks, they employ another neural network, namely an *attention mechanism* that *attends* to the entire encoder RNN states. This mechanism allows the decoder to focus on the important locations of the source sequence and use the relevant information during decoding steps for producing "better" output sequences. Recently, the concept of attention has been a popular research idea due to its capability to align different objects, e.g., in computer vision [6, 39, 40, 18] and neural machine translation [3, 19, 25]. In this study, we also employ a special attention structure for the policy parameterization. See Section 3.3 for a detailed discussion of the attention mechanism.

**Neural Combinatorial Optimization**     Over the last several years, multiple methods have been developed to tackle combinatorial optimization problems by using recent advances in artificial intelligence. The first attempt was proposed by Vinyals et al. [34], who introduce the concept of a *Pointer Network*, a model originally inspired by sequence-to-sequence models. Because it is invariant to the length of the encoder sequence, the Pointer Network enables the model to apply to combinatorial optimization problems, where the output sequence length is determined by the source sequence. They use the Pointer Network architecture in a supervised fashion to find near-optimal Traveling Salesman Problem (TSP) tours from ground truth optimal (or heuristic) solutions. This dependence on supervision prohibits the Pointer Network from finding better solutions than the ones provided during the training.

Closest to our approach, Bello et al. [4] address this issue by developing a neural combinatorial optimization framework that uses RL to optimize a policy modeled by a Pointer Network. Using several classical combinatorial optimization problems such as TSP and the knapsack problem, they show the effectiveness and generality of their architecture.

On a related topic, Dai et al. [11] solve optimization problems over graphs using a graph embedding structure [10] and a deep Q-learning (DQN) algorithm [26]. Even though VRP can be represented by a graph with weighted nodes and edges, their proposed approach does not directly apply since in VRP, a particular node (e.g. the depot) might be visited multiple times.

Next, we introduce our model, which is a simplified version of the Pointer Network.

## 3   The Model

In this section, we formally define the problem and our proposed framework for a generic combinatorial optimization problem with a given set of inputs $X \doteq \{x^i, i = 1, \cdots, M\}$. We allow some of the elements of each input to change between the decoding steps, which is, in fact, the case in many problems such as the VRP. The dynamic elements might be an artifact of the decoding procedure itself, or they can be imposed by the environment. For example, in the VRP, the remaining customer demands change over time as the vehicle visits the customer nodes; or we might consider a variant in which new customers arrive or adjust their demand values over time, independent of the vehicle decisions. Formally, we represent each input $x^i$ by a sequence of tuples $\{x_t^i \doteq (s^i, d_t^i), t = 0, 1, \cdots\}$, where $s^i$ and $d_t^i$ are the static and dynamic elements of the input, respectively, and can themselves be tuples. One can view $x_t^i$ as a vector of features that describes the state of input $i$ at time $t$. For instance, in the VRP, $x_t^i$ gives a snapshot of the customer $i$, where $s^i$ corresponds to the 2-dimensional coordinate of customer $i$'s location and $d_t^i$ is its demand at time $t$. We will denote the set of all input states at a fixed time $t$ with $X_t$.

We start from an arbitrary input in $X_0$, where we use the pointer $y_0$ to refer to that input. At every decoding time $t$ ($t = 0, 1, \cdots$), $y_{t+1}$ points to one of the available inputs $X_t$, which determines the input of the next decoder step; this process continues until a termination condition is satisfied. The termination condition is problem-specific, showing that the generated sequence satisfies the

feasibility constraints. For instance, in the VRP that we consider in this work, the terminating condition is that there is no more demand to satisfy. This process will generate a sequence of length $T$, $Y = \{y_t, t = 0, ..., T\}$, possibly with a different sequence length compared to the input length $M$. This is due to the fact that, for example, the vehicle may have to go back to the depot several times to refill. We also use the notation $Y_t$ to denote the decoded sequence up to time $t$, i.e., $Y_t = \{y_0, \cdots, y_t\}$. We are interested in finding a stochastic policy $\pi$ which generates the sequence $Y$ in a way that minimizes a loss objective while satisfying the problem constraints. The optimal policy $\pi^*$ will generate the optimal solution with probability 1. Our goal is to make $\pi$ as close to $\pi^*$ as possible. Similar to Sutskever et al. [32], we use the probability chain rule to decompose the probability of generating sequence $Y$, i.e., $P(Y|X_0)$, as follows:

$$P(Y|X_0) = \prod_{t=0}^{T} \pi(y_{t+1}|Y_t, X_t), \tag{1}$$

and

$$X_{t+1} = f(y_{t+1}, X_t) \tag{2}$$

is a recursive update of the problem representation with the state transition function $f$. Each component in the right-hand side of (1) is computed by the attention mechanism, i.e.,

$$\pi(\cdot|Y_t, X_t) = \text{softmax}(g(h_t, X_t)), \tag{3}$$

where $g$ is an affine function that outputs an input-sized vector, and $h_t$ is the state of the RNN decoder that summarizes the information of previously decoded steps $y_0, \cdots, y_t$. We will describe the details of our proposed attention mechanism in Section 3.3.

**Remark 1**: This structure can handle combinatorial optimization problems in both a more classical static setting as well as in dynamically changing ones. In static combinatorial optimization, $X_0$ fully defines the problem that we are trying to solve. For example, in the VRP, $X_0$ includes all customer locations as well as their demands, and the depot location; then, the remaining demands are updated with respect to the vehicle destination and its load. With this consideration, often there exists a well-defined Markovian transition function $f$, as defined in (2), which is sufficient to update the dynamics between decision points. However, our framework can also be applied to problems in which the state transition function is unknown and/or is subject to external noise, since the training does not explicitly make use of the transition function. However, knowing this transition function helps in simulating the environment that the training algorithm interacts with. See Appendix C.6 for an example of how to handle a stochastic version of the VRP in which random customers with random demands appear over time.

## 3.1 Limitations of Pointer Networks

Although the framework proposed by Bello et al. [4] works well on problems such as the knapsack problem and TSP, it is not efficient to more complicated combinatorial optimization problems in which the system representation varies over time, such as VRP. Bello et al. [4] feed a random sequence of inputs to the RNN encoder. Figure 1 illustrates with an example why using the RNN in the encoder is restrictive. Suppose that at the first decision step, the policy sends the vehicle to customer 1, and as a result, its demand is satisfied, i.e., $d_0^1 \neq d_1^1$. Then in the second decision step, we need to re-calculate the whole network with the new $d_1^1$ information in order to choose the next customer. The dynamic elements complicate the forward pass of the network since there should be encoder/decoder updates when an input changes. The situation is even worse during back-propagation to accumulate the gradients since we need to remember when the dynamic elements changed. In order to resolve this complication, we require the policy model to be *invariant to the input sequence* so that changing the order of any two inputs does not affect the network. In Section 3.2, we present a simple network that satisfies this property.

## 3.2 The Proposed Neural Network Model

We argue that the RNN encoder adds extra complication to the encoder but is actually not necessary, and the approach can be made much more general by omitting it. RNNs are necessary only when the inputs convey sequential information; e.g., in text translation the combination of words and their

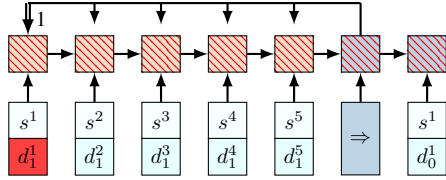

Figure 1: Limitation of the Pointer Network. After a change in dynamic elements ($d_1^1$ in this example), the whole Pointer Network must be updated to compute the probabilities in the next decision point.

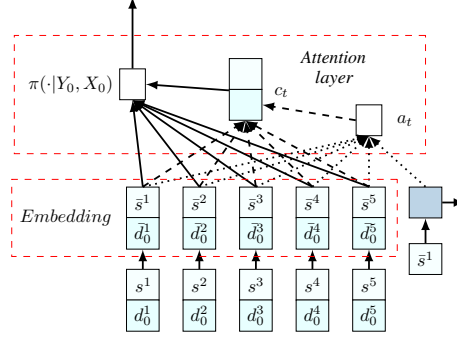

Figure 2: Our proposed model. The embedding layer maps the inputs to a high-dimensional vector space. On the right, an RNN decoder stores the information of the decoded sequence. Then, the RNN hidden state and embedded input produce a probability distribution over the next input using the attention mechanism.

relative position must be captured in order for the translation to be accurate. But the question here is, *why do we need to have them in the encoder for combinatorial optimization problems when there is no meaningful order in the input set?* As an example, in the VRP, the inputs are the set of unordered customer locations with their respective demands, and their order is not meaningful; any random permutation contains the same information as the original inputs. Therefore, in our model, we simply leave out the encoder RNN and directly use the embedded inputs instead of the RNN hidden states. By this modification, many of the computational complications disappear, without decreasing the efficiency. In Appendix A, we provide experiments to verify this claim.

As illustrated in Figure 2, our model is composed of two main components. The first is a set of graph embeddings [30] that can be used to encode structured data inputs. Among the available techniques, we tried a one-layer Graph Convolutional Network [21] embedding, but it did not show any improvement on the results, so we kept the embedding in this paper simple by utilizing the local information at each node, e.g., its coordinates and demand values, without incorporating adjacency information. In fact, this embeddings maps each customer's information into a $D$-dimensional vector space encoding. We might have multiple embeddings corresponding to different elements of the input, but they are shared among the inputs. The second component is a decoder that points to an input at every decoding step. As is common in the literature [3, 32, 7], we use RNN to model the decoder network. Notice that we feed the static elements as the inputs to the decoder network. The dynamic element can also be an input to the decoder, but our experiments on the VRP do not suggest any improvement by doing so. For this reason, the dynamic elements are used only in the attention layer, described next.

### 3.3 Attention Mechanism

An attention mechanism is a differentiable structure for addressing different parts of the input. Figure 2 illustrates the attention mechanism employed in our method. At decoder step $i$, we utilize a context-based attention mechanism with glimpse, similar to Vinyals et al. [35], which extracts the relevant information from the inputs using a variable-length alignment vector $a_t$. In other words, $a_t$ specifies how much every input data point might be relevant in the next decoding step $t$.

Let $\bar{x}_t^i = (\bar{s}^i, \bar{d}_t^i)$ be the embedded input $i$, and $h_t \in \mathbb{R}^D$ be the memory state of the RNN cell at decoding step $t$. The alignment vector $a_t$ is then computed as

$$a_t = a_t(\bar{x}_t, h_t) = \text{softmax}(u_t), \quad \text{where } u_t^i = v_a^T \tanh\left(W_a[\bar{x}_t^i; h_t]\right). \quad (4)$$

Here ";" means the concatenation of two vectors. We compute the conditional probabilities by combining the context vector $c_t$, computed as

$$c_t = \sum_{i=1}^{M} a_t^i \bar{x}_t^i, \quad (5)$$

with the embedded inputs, and then normalizing the values with the softmax function, as follows:

$$\pi(\cdot|Y_t, X_t) = \text{softmax}(\tilde{u}_t), \quad \text{where } \tilde{u}_t^i = v_c^T \tanh\left(W_c[\bar{x}_t^i; c_t]\right). \tag{6}$$

In (4)–(6), $v_a$, $v_c$, $W_a$ and $W_c$ are trainable variables.

**Remark 2**: *Model Symmetry*: Vinyals et al. [35] discuss an extension of sequence-to-sequence models where they empirically demonstrate that in tasks with no obvious input sequence, such as sorting, the order in which the inputs are fed into the network matter. A similar concern arises when using Pointer Networks for combinatorial optimization problems. However, the policy model proposed in this paper does not suffer from such a complication since the embeddings and the attention mechanism are invariant to the input order.

## 3.4 Training Method

To train the network, we use well-known policy gradient approaches. To use these methods, we parameterize the stochastic policy $\pi$ with parameters $\theta$, where $\theta$ is vector of all trainable variables used in embedding, decoder, and attention mechanism. Policy gradient methods use an estimate of the gradient of the expected return with respect to the policy parameters to iteratively improve the policy. In principle, the policy gradient algorithm contains two networks: (*i*) an actor network that predicts a probability distribution over the next action at any given decision step, and (*ii*) a critic network that estimates the reward for any problem instance from a given state. Our training methods are quite standard, and due to space limitation we leave the details to the Appendix.

## 4   Computational Experiment

Many variants of the VRP have been extensively studied in the operations research literature. See, for example, the reviews by Laporte [23], Laporte et al. [24], or the book by Toth and Vigo [33] for different variants of the problem. In this section, we consider a specific capacitated version of the problem in which one vehicle with a limited capacity is responsible for delivering items to many geographically distributed customers with finite demands. When the vehicle's load runs out, it returns to the depot to refill. We will denote the vehicle's remaining load at time $t$ as $l_t$. The objective is to minimize the total route length while satisfying all of the customer demands. This problem is often called the capacitated VRP (CVRP) to distinguish it from other variants, but we will refer to it simply as the VRP.

We assume that the node locations and demands are randomly generated from a fixed distribution. Specifically, the customers and depot locations are randomly generated in the unit square $[0, 1] \times [0, 1]$. For simplicity of exposition, we assume that the demand of each node is a discrete number in $\{1, .., 9\}$, chosen uniformly at random. We note, however, that the demand values can be generated from any distribution, including continuous ones.

We assume that the vehicle is located at the depot at time 0, so the first input to the decoder is an embedding of the depot location. At each decoding step, the vehicle chooses from among the customer nodes or the depot to visit in the next step. After visiting customer node $i$, the demands and vehicle load are updated as follows:

$$d_{t+1}^i = \max(0, d_t^i - l_t), \quad d_{t+1}^k = d_t^k \text{ for } k \neq i, \text{and} \quad l_{t+1} = \max(0, l_t - d_t^i) \tag{7}$$

which is an explicit definition of the state transition function (2) for the VRP. Once a sequence of the nodes to be visited is sampled, we compute the total vehicle distance and use its negative value as the reward signal.

In this experiment, we have employed two different decoders: *(i)* greedy, in which at every decoding step, the node (either customer or depot) with the highest probability is selected as the next destination, and *(ii)* beam search (BS), which keeps track of the most probable paths and then chooses the one with the minimum tour length [28]. Our results indicate that by applying the beam search algorithm, the quality of the solutions can be improved with only a slight increase in computation time.

For faster training and generating feasible solutions, we have used a *masking scheme* which sets the log-probabilities of infeasible solutions to $-\infty$ or forces a solution if a particular condition is satisfied. In the VRP, we use the following masking procedures: *(i)* nodes with zero demand are not allowed to be visited; *(ii)* all customer nodes will be masked if the vehicle's remaining load is

exactly 0; and *(iii)* the customers whose demands are greater than the current vehicle load are masked. Notice that under this masking scheme, the vehicle must satisfy all of a customer's demands when visiting it. We note, however, that if the situation being modeled does allow split deliveries, one can relax *(iii)*. Indeed, the relaxed masking allows split deliveries, so the solution can allocate the demands of a given customer into multiple routes. This property is, in fact, an appealing behavior that is present in many real-world applications, but is often neglected in classical VRP algorithms. In all the experiments of the next section, we do not allow to split demands. Further investigation and illustrations of this property is included in Appendix C.2–C.4.

## 4.1 Results

In this section, we compare the solutions found using our framework with those obtained from the *Clarke-Wright savings heuristic* (CW), the *Sweep heuristic* (SW), and Google's optimization tools (OR-Tools). We run our tests on problems with 10, 20, 50 and 100 customer nodes and corresponding vehicle capacity of 20, 30, 40 and 50; for example, VRP10 consists of 10 customer and the default vehicle capacity is 20 unless otherwise specified. The results are based on 1000 instances, sampled for each problem size.

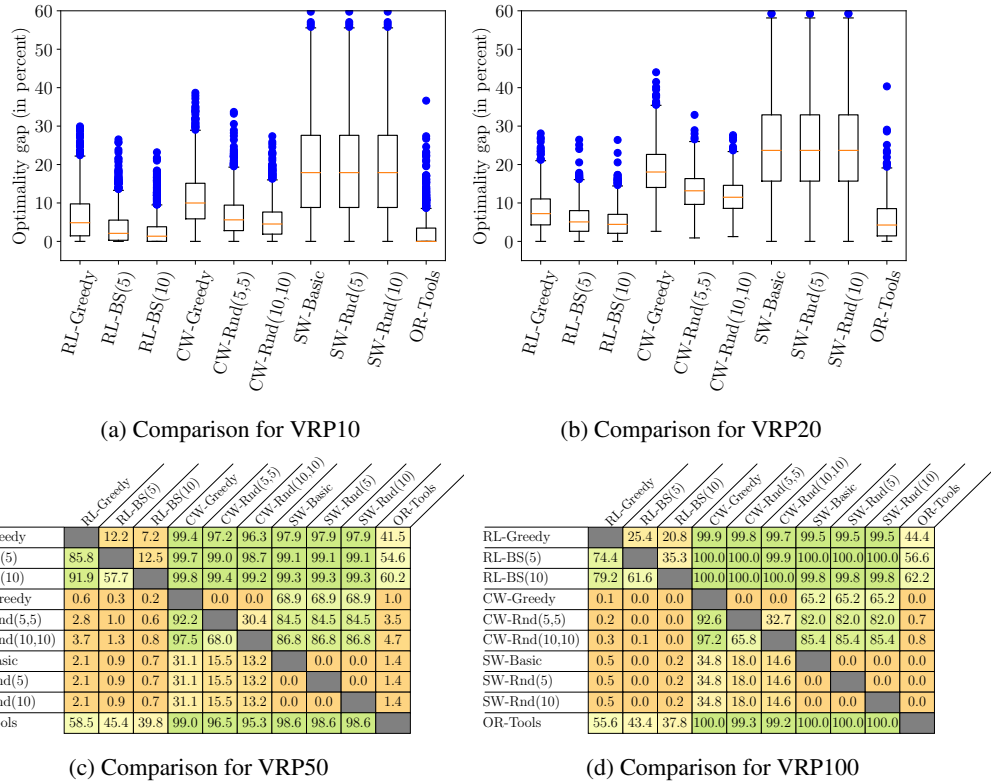

(a) Comparison for VRP10        (b) Comparison for VRP20

(c) Comparison for VRP50        (d) Comparison for VRP100

Figure 3: Parts 3a and 3b show the optimality gap (in percent) using different algorithms/solvers for VRP10 and VRP20. Parts 3c and 3d give the proportion of the samples for which the algorithms in the rows outperform those in the columns; for example, RL-BS(5) is superior to RL-greedy in 85.8% of the VRP50 instances.

Figure 3 shows the distribution of total tour lengths generated by our method, using greedy and BS decoders, with the number inside the parentheses indicating the beam-width parameter. In the experiments, we label our method with the "RL" prefix. In addition, we also implemented a randomized version of both heuristic algorithms to improve the solution quality; for Clarke-Wright, the numbers inside the parentheses are the randomization depth and randomization iterations parameters; and for Sweep, it is the number of random initial angles for grouping the nodes. Finally, we use Google's OR-Tools [16], which is a more competitive baseline. See Appendix B for a detailed discussion on the baselines.

For small problems of VRP10 and VRP20, it is possible to find the optimal solution, which we do by solving a mixed integer formulation of the VRP [33]. Figures 3a and 3b measure how far the solutions are far from optimality. The optimality gap is defined as the distance from the optimal objective value normalized by the latter. We observe that using a beam width of 10 is the best-performing method; roughly 95% of the instances are at most 10% away from optimality for VRP10 and 13% for VRP20. Even the outliers are within 20–25% of optimality, suggesting that our RL-BS methods are robust in comparison to the other baseline approaches.

Since obtaining the optimal objective values for VRP50 and VRP100 is not computationally affordable, in Figures 3d and 3d, we compare the algorithms in terms of their winning rate. Each table gives the percentage of the instances in which the algorithms in the rows outperform those in the columns. In other words, the cell corresponding to (A,B) shows the percentage of the samples in which algorithm A provides shorter tours than B. We observe that the classical heuristics are outperformed by the other approaches in almost 100% of the samples. Moreover, RL-greedy is comparable with OR-Tools, but incorporating beam search into our framework increases the winning rate of our approach to above 60%.

Figure 4 shows the solution times normalized by the number of customer nodes. We observe that this ratio stays almost the same for RL with different decoders. In contrast, the run time for the Clarke-Wright and Sweep heuristics increases faster than linearly with the number of nodes. This observation is one motivation for applying our framework to more general combinatorial problems, since it suggests that our method scales well. Even though the greedy Clark-Wright and basic Sweep heuristics are fast for small instances, they do not provide competitive solutions. Moreover, for larger problems, our framework is faster than the randomized heuristics. We also include the solution times for OR-Tools in the graph, but we should note that OR-Tools is implemented in C++, which makes exact time comparisons impossible since the other baselines are implemented in Python. It is worthwhile to mention that the runtimes reported for the RL methods are for the case when we decode a single problem at a time. It is also possible to decode all 1000 test problems in a batch which will result in approximately $50\times$ speed up. For example, one-by-one decoding of VRP10 for 1000 instances takes around 50 seconds, but by passing all 1000 instances to decoder at once, the total decoding time decreases to around 1 second on a K80 GPU.

Active search is another method used by [4] to assist with the RL training on a specific problem instance in order to iteratively search for a better solution. We do not believe that active search is a practical tool for our problem. One reason is that it is very time-consuming. A second is that we intend to provide a solver that produces solutions by just scoring a trained policy, while active search requires a separate training phase for every instance. To test our conjecture that active search will not be effective for our problem, we implemented active search for VRP10 with samples of size 640 and 1280, and the average route length was 4.78 and 4.77 with 15s and 31s solution times per instance, which are far worse than BS decoders. Note that BS(5) and BS(10) give 4.72 and 4.68, in less than 0.1s. For this reason, we exclude active search from our comparisons.

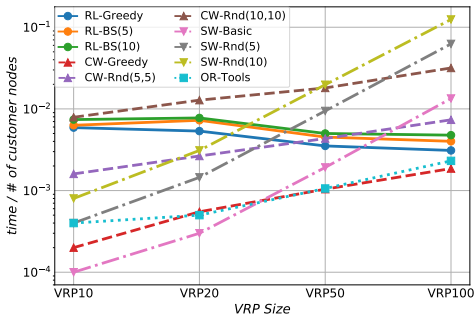

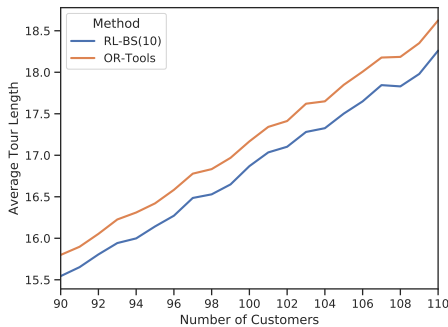

Figure 4: Ratio of solution time to the number of customer nodes using different algorithms.

Figure 5: Trained for VRP100 and tested for VRP90-VRP110.

One desired property of the method is that it should be able to handle variable problem sizes. To test this property, we designed two experiments. In the first experiment, we used the trained policy for VRP100 and evaluated its performance on VRP90-VRP110. As it can be seen in Figure 5, our method with BS(10) outperforms OR-Tools on all problem sizes. In the second experiment, we test

the generalization when the problems are very different. More specifically, we use the models trained for VRP50-Cap40 and VRP50-Cap50 in order to generate a solution for VRP100-Cap50. Using BS(10), the average tour length is 18.00 and 17.80, which is still better than the classical heuristics, but worse than OR-Tools. Overall, these two experiments suggest that when the problems are close in terms of the number of customer and vehicle capacity, it is reasonable to expect a near-optimal solution, but we will see a degradation when the testing problems are very different from the training ones.

## 4.2    Extension to Other VRPs

The proposed framework can be extended easily to problems with multiple depots; one only needs to construct the corresponding state transition function and masking procedure. It is also possible to include various side constraints: soft constraints can be applied by penalizing the rewards, or hard constraints such as time windows can be enforced through a masking scheme. However, designing such a scheme might be a challenging task, possibly harder than solving the optimization problem itself. Another interesting extension is for VRPs with multiple vehicles. In the simplest case in which the vehicles travel independently, one must only design a shared masking scheme to avoid the vehicles pointing to the same customer nodes. Incorporating competition or collaboration among the vehicles is also an interesting line of research that relates to multi-agent RL (MARL) [5].

This framework can also be applied to real-time services including on-demand deliveries and taxis. In Appendix C.6, we design an experiment to illustrate the performance of the algorithm on a VRP where both customer locations and their demands are subject to change. Our results indicate superior performance than the baselines.

## 5    Discussion and Conclusion

According to the findings of this paper, our RL algorithm is competitive with state-of-the-art VRP heuristics, and this represents progress toward solving the VRP with RL for real applications. The fact that we can solve similar-sized instances without retraining for every new instance makes it easy to deploy our method in practice. For example, a vehicle equipped with a processor can use the trained model and solve its own VRP, only by doing a sequence of pre-defined arithmetic operations. Moreover, unlike many classical heuristics, our proposed method scales well as the problem size increases, and it has superior performance with competitive solution-time. It does not require a distance matrix calculation, which might be computationally cumbersome, especially in dynamically changing VRPs. One important discrepancy which is usually neglected by the classical heuristics is that one or more of the elements of the VRP are stochastic in the real world. In this paper, we also illustrate that the proposed RL-based method can be applied to a more complicated stochastic version of the VRP. In summary, we expect that the proposed architecture has significant potential to be used in real-world problems with further improvements and extensions that incorporate other realistic constraints.

Noting that the proposed algorithm is not limited to VRP, it will be an important topic of future research to apply it to other combinatorial optimization problems such as bin-packing and job-shop or flow-shop scheduling. This method is quite appealing since the only requirement is a verifier to find feasible solutions and also a reward signal to demonstrate how well the policy is working. Once the trained policy is available, it can be used many times, without needing to re-train for new problems as long as they are generated from the training distribution.

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
