[Supplementary Material · vrp_neurips_2018_supplementary.pdf]

# A Our Policy Model versus Pointer Network

In the first experiment of this section, we use the *Traveling Salesman Problem* (TSP) (a special case of the VRP in which there is only a single route to optimize) as the test-bed to validate the performance of the proposed method. We compare the route lengths of the TSP solutions obtained by our framework with those given by the model of Bello et al. [4] for random instances with 20, 50, and 100 nodes. In the training phase, we generate $10^6$ TSP instances for each problem size, and use them in training for 20 epochs. $10^6$ is chosen because we want to have a diverse set of problem configurations; it can be larger or smaller, or we can generate the instances on-the-fly as long as we make sure that the instance are drawn from the same probability distribution with the same random seed. The city locations are chosen uniformly from the unit square $[0, 1] \times [0, 1]$. We use the same data distribution to produce instances for the testing phase. The decoding process starts from a random TSP node and the termination criterion is that all cities are visited. We also use a masking scheme to prohibit visiting nodes more than once.

Table 1 summarizes the results for different TSP sizes using the *greedy decoder* in which at every decoding step, the city with the highest probability is chosen as the destination. The results are averaged over 1000 instances. The first column is the average TSP tour length using our proposed architecture, the second column is the result of our implementation of Bello et al. [4] with greedy decoder, and the optimal tour lengths are reported in the last column. To obtain the optimal values, we solved the TSP using the Concorde optimization software [1]. A comparison of the first two columns suggests that there is almost no difference between the performance of our framework and Pointer-RL. In fact, the RNN encoder of the Pointer Network learns to convey no information to the next steps, i.e., $h_t = f(x_t)$. On the other hand, our approach is around 60% faster in both training and inference, since it has two fewer RNNs—one in the encoder of actor network and another in the encoder of critic network. Table 1 also summarizes the training times for one epoch of the training and the time-savings that we gain by eliminating the encoder RNNs.

Table 1: Average tour length for TSP and training time for one epoch (in minutes).

| | Average tour length | | | Training time | | |
|---|---|---|---|---|---|---|
| Task | Our Framework (Greedy) | Pointer-RL (Greedy) | Optimal | Our Framework (Greedy) | Pointer-RL (Greedy) | % Time Saving |
| TSP20 | 3.97 | 3.96 | 3.84 | 22.18 | 50.33 | 55.9% |
| TSP50 | 6.08 | 6.05 | 5.70 | 54.10 | 147.25 | 63.3% |
| TSP100 | 8.44 | 8.45 | 7.77 | 122.10 | 300.73 | 59.4% |

Based on the discussion in section 3.1, the main problem with applying Pointer Networks is mainly computational, and in the next experiment of this section, we compare the learning process of our model with that of Pointer Networks. We implemented a Pointer Network for VRP10, and as is illustrated in Figure 6, its performance is much worse, and each training epoch takes around $2.5\times$ longer to train.

Figure 6: Comparison with Pointer Network for VRP

# B Capacitated VRP Baselines

In this Appendix, we briefly describe the algorithms and solvers that we used as benchmarks. More details and examples of these algorithms can be found in Snyder and Shen [31]. The first two baseline approaches are well-known heuristics designed for VRP. Our third baseline is Google's optimization tools, which includes one of the best open-source VRP solvers. Finally, we compute the optimal solutions for small VRP instances, so we can measure how far the solutions are from optimality.

## B.1 Clarke-Wright Savings Heuristic

The Clarke-Wright savings heuristic [9] is one of the best-known heuristics for the VRP. Let $\mathcal{N} \doteq \{1, \cdots, N\}$ be the set of customer nodes, and 0 be the depot. The distance between nodes $i$ and $j$ is denoted by $c_{ij}$, and $c_{0i}$ is the distance of customer $i$ from the depot. Algorithm 1 describes a randomized version of the heuristic. The basic idea behind this algorithm is that it initially considers a separate route for each customer node $i$, and then reduces the total cost by iteratively merging the routes. Merging two routes by adding the edge $(i, j)$ reduces the total distance by $s_{ij} = c_{i0} + c_{0j} - c_{ij}$, so the algorithm prefers mergers with the highest savings $s_{ij}$.

We introduce two hyper-parameters, $R$ and $M$, which we refer to as the *randomization depth* and *randomization iteration*, respectively. When $M = R = 1$, this algorithm is equivalent to the original Clarke-Wright savings heuristic, in which case, the feasible merger with the highest savings will be selected. By allowing $M, R > 1$, we introduce randomization, which can improve the performance of the algorithm further. In particular, Algorithm 1 chooses randomly from the $r \in \{1, \cdots, R\}$ best feasible mergers. Then, for each $r$, it solves the problem $m \in \{1, \cdots, M\}$ times, and returns the solution with the shortest total distance.

---
**Algorithm 1** Randomized Clarke-Wright Savings Heuristic

---
1: compute savings $s_{ij}$, where
$$s_{ij} = c_{i0} + c_{0j} - c_{ij} \qquad\qquad i, j \in \mathcal{N}, i \neq j$$
$$s_{ii} = 0 \qquad\qquad i \in \mathcal{N}$$
2: **for** $r = 1, \cdots, R$ **do**
3:     **for** $m = 1, \cdots, M$ **do**
4:         place each $i \in \mathcal{N}$ in its own route
5:         **repeat**
6:             find $k$ feasible mergers $(i, j)$ with the highest $s_{ij} > 0$, satisfying the following conditions:
               i) $i$ and $j$ are in different routes
               ii) both $i$ and $j$ are adjacent to the depot
               iii) combined demand of routes containing $i$ and $j$ is $\leq$ vehicle capacity
7:             choose a random $(i, j)$ from the feasible mergers, and combine the associated routes by replacing $(i, 0)$ and $(0, j)$ with $(i, j)$
8:         **until** no feasible merger is left
9:     **end for**
10: **end for**
11: **Return**: route with the shortest length

---

## B.2 Sweep Heuristic

The sweep heuristic [38] solves the VRP by breaking it into multiple TSPs. By rotating an arc emanating from the depot, it groups the nodes into several clusters, while ensuring that the total demand of each cluster is not violating the vehicle capacity. Each cluster corresponds to a TSP that can be solved by using an exact or approximate algorithm. In our experiments, we use dynamic programming to find the optimal TSP tour. After solving TSPs, the VRP solution can be obtained by combining the TSP tours. Algorithm 2 shows the pseudo-code of this algorithm.

## B.3 Google's OR-Tools

Google Optimization Tools (OR-Tools) [16] is an open-source solver for combinatorial optimization problems. OR-Tools contains one of the best available VRP solvers, which has implemented many

---
**Algorithm 2** Randomized Sweep Algorithm
---
1: for each $i \in \mathcal{N}$, compute angle $\alpha_i$, respective to depot location
2: $l \leftarrow$ vehicle capacity
3: **for** $r = 1, \cdots, R$ **do**
4:     select a random angle $\alpha$
5:     $k \leftarrow 0$; initialize cluster $S_k \leftarrow \emptyset$
6:     **repeat**
7:         increase $\alpha$ until it equal to some $\alpha_i$
8:         **if** demand $d_i > l$ **then**
9:             $k \leftarrow k + 1$
10:             $S_k \leftarrow \emptyset$
11:             $l \leftarrow$ vehicle capacity
12:         **end if**
13:         $S_k \leftarrow S_k \cup \{i\}$
14:         $l \leftarrow l - d_i$
15:     **until** no unclustered node is left
16:     solve a TSP for each $S_k$
17:     merge TSP tours to produce a VRP route
18: **end for**
19: **Return**: route with the shortest length
---

heuristics (e.g., Clarke-Wright savings heuristic [9], Sweep heuristic [38], Christofides' heuristic [8] and a few others) for finding an initial solution and metaheuristics (e.g. Guided Local Search [36], Tabu Search [14] and Simulated Annealing [22]) for escaping from local minima in the search for the best solution. The default version of the OR-Tools VRP solver does not exactly match the VRP studied in this paper, but with a few adjustments, we can use it as our baseline. The first limitation is that OR-Tools only accepts integer locations for the customers and depot while our problem is defined on the unit square. To handle this issue, we scale up the problem by multiplying all locations by $10^4$ (meaning that we will have 4 decimal digits of precision), so the redefined problem is now in $[0, 10^4] \times [0, 10^4]$. After solving the problem, we scale down the solutions and tours to get the results for the original problem. The second difference is that OR-Tools is defined for a VRP with multiple vehicles, each of which can have at most one tour. One can verify that by setting a large number of vehicles (10 in our experiments), it is mathematically equivalent to our version of the VRP.

### B.4 Optimal Solution

We use a mixed integer formulation for the VRP [33] and the Gurobi optimization solver [17] to obtain the optimal VRP tours. VRP has an exponential number of constraints, and of course, it requires careful tricks for even small problems. In our implementation, we start off with a relaxation of the capacity constraints and solve the resulting problem to obtain a lower bound on the optimal objective value. Then we check the generated tours and add the capacity constraint as *lazy-constraints* if a specific subtour's demand has violated the vehicle capacity, or the subtour does not include the depot. With this approach, we were able to find the optimal solutions for VRP10 and VRP20, but this method is intractable for larger VRPs; for example, on a single instance of VRP50, the solution has 6.7% optimality gap after 10000 seconds.

## C  Extended Results of the VRP Experiment

In this section, we present more detailed results for the VRP, including a comparison with baselines and an illustration of the solutions generated. We demonstrate the flexibility of the framework to incorporate split deliveries, as an option, to further improve the solution quality. We also illustrate with an example that our proposed framework can be applied to more challenging VRPs with the stochastic elements.

## C.1 Implementation Details

For the embedding, we use 1-dimensional convolution layers for the embedding, in which the in-width is the input length, the number of filters is $D$, and the number of in-channels is the number of elements of $x$. We find that training without an embedding layer always yields an inferior solution. One possible explanation is that the policy is able to extract useful features from the high-dimensional input representations much more efficiently. Recall that our embedding is an affine transformation, so it does not necessarily keep the embedded input distances proportional to the original 2-dimensional Euclidean distances.

We use one layer of LSTM RNN in the decoder with a state size of 128. Each customer location is also embedded into a vector of size 128, shared among the inputs. We employ similar embeddings for the dynamic elements; the demand $d_t^i$ and the remaining vehicle load after visiting node $i$, $l_t - d_t^i$, are mapped to a vector in a 128-dimensional vector space and used in the attention layer. In the critic network, first, we use the output probabilities of the actor network to compute a weighted sum of the embedded inputs, and then, it has two hidden layers: one dense layer with ReLU activation and another linear one with a single output. The variables in both actor and critic network are initialized with Xavier initialization [13]. For training both networks, we use the REINFORCE Algorithm and Adam optimizer [20] with learning rate $10^{-4}$. The batch size $N$ is 128, and we clip the gradients when their norm is greater than 2. We use dropout with probability 0.1 in the decoder LSTM. Moreover, we tried the entropy regularizer [37, 27], which has been shown to be useful in preventing the algorithm from getting stuck in local optima, but it does not show any improvement in our experiments; therefore, we do not use it in the results reported in this paper.

On a single GPU K80, every 100 training steps of the VRP with 20 customer nodes takes approximately 35 seconds. Training for 20 epochs requires about 13.5 hours. The TensorFlow implementation of our code is available at `https://github.com/OptMLGroup/VRP-RL`.

## C.2 Flexibility to VRPs with Split Demands

In the classical VRP that we studied in Section 4, each customer is required to be visited exactly once. On the contrary to what is usually assumed in the classical VRP, one can relax this constraint to obtain savings by allowing split deliveries [2]. In this section, we show that this relaxation is straightforward by slightly modifying the masking scheme. Basically, we omit the condition *(iii)* from the masking introduced in Section 4, and use the new masking with the exactly similar model; we want to emphasize that we do not re-train the policy model and use the variables trained previously, so this property is achieved at no extra cost.

Figure 7 shows the improvement by relaxing these constraints, where we label our relaxed method with "RL-SD". Other heuristics does not have such option and they are reported for the original (not relaxed) problem. In parts 7a and 7b we illustrate the "optimality" gap for VRP10 and VRP20, respectively. What we refer to optimality in this section (and other places in this paper) is the optimal objective value of the non-relaxed problem. Of course, the relaxed problem would have a lower optimal objective value. That is why RL-SD obtains negative values in these plots. We see that RL-SD can effectively use split delivery to obtain solutions that are around $5-10\%$ shorter than the "optimal" tours. Similar to 3, parts 7c and 7d show the winning percentage of the algorithms in rows in comparison to the ones in columns. We observe that the winning percentage of RL-SD methods significantly improves after allowing the split demands. For example in VRP50 and VRP100, RL-SD-Greedy is providing competitive results with OR-Tools, or RL-SD-BS(10) outperforms OR-Tools in roughly $67\%$ of the instances, while this number was around $61\%$ before relaxation.

## C.3 Summary of Comparison with Baselines

Table 2 provides the average and the standard deviation of tour lengths for different VRPs. We also test the RL approach using the split delivery option where the customer demands can be satisfied in more than one subtours (labeled with "RL-SD", at the end of the table). We observe that the average total length of the solutions found by our method using various decoders outperforms the heuristic algorithms and OR-Tools. We also see that using the beam search decoder significantly improves the solution while only adding a small computational cost in run-time. Also allowing split delivery enables our RL-based methods to improve the total tour length by a factor of around $0.6\%$ on average. We also present the solution time comparisons in this table, where all the times are reported on a

(a) Comparison for VRP10          (b) Comparison for VRP20

(c) Comparison for VRP50

| | RL-SD-Greedy | RL-SD-BS(5) | RL-SD-BS(10) | CW-Greedy | CW-Rnd(5,5) | CW-Rnd(10,10) | SW-Basic | SW-Rnd(5) | SW-Rnd(10) | OR-Tools |
|---|---|---|---|---|---|---|---|---|---|---|
| RL-SD-Greedy | | 15.8 | 8.7 | 99.7 | 98.0 | 96.9 | 98.8 | 98.8 | 98.8 | 46.8 |
| RL-SD-BS(5) | 82.1 | | 15.4 | 99.7 | 99.4 | 99.3 | 99.4 | 99.4 | 99.4 | 60.5 |
| RL-SD-BS(10) | 90.4 | 59.2 | | 99.9 | 99.5 | 99.6 | 99.6 | 99.6 | 99.6 | 66.0 |
| CW-Greedy | 0.3 | 0.3 | 0.1 | | 0.0 | 0.0 | 68.9 | 68.9 | 68.9 | 1.0 |
| CW-Rnd(5,5) | 2.0 | 0.6 | 0.5 | 92.2 | | 30.4 | 84.5 | 84.5 | 84.5 | 3.5 |
| CW-Rnd(10,10) | 3.1 | 0.7 | 0.4 | 97.5 | 68.0 | | 86.8 | 86.8 | 86.8 | 4.7 |
| SW-Basic | 1.2 | 0.6 | 0.4 | 31.1 | 15.5 | 13.2 | | 0.0 | 0.0 | 1.4 |
| SW-Rnd(5) | 1.2 | 0.6 | 0.4 | 31.1 | 15.5 | 13.2 | 0.0 | | 0.0 | 1.4 |
| SW-Rnd(10) | 1.2 | 0.6 | 0.4 | 31.1 | 15.5 | 13.2 | 0.0 | 0.0 | | 1.4 |
| OR-Tools | 53.2 | 39.5 | 34.0 | 99.0 | 96.5 | 95.3 | 98.6 | 98.6 | 98.6 | |

(d) Comparison for VRP100

| | RL-SD-Greedy | RL-SD-BS(5) | RL-SD-BS(10) | CW-Greedy | CW-Rnd(5,5) | CW-Rnd(10,10) | SW-Basic | SW-Rnd(5) | SW-Rnd(10) | OR-Tools |
|---|---|---|---|---|---|---|---|---|---|---|
| RL-SD-Greedy | | 27.5 | 20.3 | 100.0 | 100.0 | 100.0 | 99.9 | 99.9 | 99.9 | 50.9 |
| RL-SD-BS(5) | 72.3 | | 36.4 | 100.0 | 100.0 | 100.0 | 99.9 | 99.9 | 99.9 | 63.5 |
| RL-SD-BS(10) | 79.7 | 61.7 | | 100.0 | 100.0 | 100.0 | 100.0 | 100.0 | 100.0 | 68.6 |
| CW-Greedy | 0.0 | 0.0 | 0.0 | | 0.0 | 0.0 | 65.2 | 65.2 | 65.2 | 0.0 |
| CW-Rnd(5,5) | 0.0 | 0.0 | 0.0 | 92.6 | | 32.7 | 82.0 | 82.0 | 82.0 | 0.7 |
| CW-Rnd(10,10) | 0.0 | 0.0 | 0.0 | 97.2 | 65.8 | | 85.4 | 85.4 | 85.4 | 0.8 |
| SW-Basic | 0.1 | 0.1 | 0.0 | 34.8 | 18.0 | 14.6 | | 0.0 | 0.0 | 0.0 |
| SW-Rnd(5) | 0.1 | 0.1 | 0.0 | 34.8 | 18.0 | 14.6 | 0.0 | | 0.0 | 0.0 |
| SW-Rnd(10) | 0.1 | 0.1 | 0.0 | 34.8 | 18.0 | 14.6 | 0.0 | 0.0 | | 0.0 |
| OR-Tools | 49.1 | 36.5 | 31.4 | 100.0 | 99.3 | 99.2 | 100.0 | 100.0 | 100.0 | |

Figure 7: Parts 3a and 3b show the "optimality" gap (in percent) using different algorithms/solvers for VRP10 and VRP20. Parts 3c and 3d give the proportion of the samples (in percent) for which the algorithms in the rows outperform those in the columns; for example, RL-BS(5) is provides shorter tours compared to RL-greedy in 82.1% of the VRP50 instances.

single core Intel 2.6 GHz CPU. It is worthwhile to mention that, unlike other RL areas, our findings are not affected by the training seed. This is because during the training, we generate $10^6$ problem instances, which is quite adequate to cover various realizations of the problem, and changing the random seed does not significantly change the training and testing instances.

## C.4 Sample VRP Solutions

Figure 8 illustrates sample VRP20 and VRP50 instances decoded by the trained model. The greedy and beam-search decoders were used to produce the figures in the top and bottom rows, respectively. It is evident that these solutions are not optimal. For example, in part (a), one of the routes crosses itself, which is never optimal in Euclidean VRP instances. Another similar suboptimality is evident in part (c) to make the total distance shorter. However, the figures illustrate how well the policy model has understood the problem structure. It tries to satisfy demands at nearby customer nodes until the vehicle load is small. Then, it automatically comprehends that visiting further nodes is not the best decision, so it returns to the depot and starts a new tour. One interesting behavior that the algorithm has learned can be seen in part (c), in which the solution reduces the cost by making a partial delivery; in this example, we observe that the red and blue tours share a customer node with demand 8, each satisfying a portion of its demand; in this way, we are able to meet all demands without needing to initiate a new tour. We also observe how using the beam-search decoder produces further improvements; for example, as seen in parts (b)–(c), it reduces the number of times when a tour crosses itself; or it reduces the number of tours required to satisfy all demands as is illustrated in (b).

Tables 3 and 4 present the RL solutions using the greedy and beam search decoders for two sample VRP10 instances with a vehicle capacity of 20. We have 10 customers indexed $0 \cdots 9$ and the location with the index 10 corresponds to the depot. The first line specifies the customer locations as well

Table 2: Average tour length, standard deviations of the tours and the average solution time (in seconds) using different baselines over a test set of size 1000.

| Baseline | VRP10, Cap20 | | | VRP20, Cap30 | | | VRP50, Cap40 | | | VRP100, Cap50 | | |
|---|---|---|---|---|---|---|---|---|---|---|---|---|
| | mean | std | time | mean | std | time | mean | std | time | mean | std | time |
| RL-Greedy | 4.84 | 0.85 | 0.049 | 6.59 | 0.89 | 0.105 | 11.39 | 1.31 | 0.156 | 17.23 | 1.97 | 0.321 |
| RL-BS(5) | 4.72 | 0.83 | 0.061 | 6.45 | 0.87 | 0.135 | 11.22 | 1.29 | 0.208 | 17.04 | 1.93 | 0.390 |
| RL-BS(10) | 4.68 | 0.82 | 0.072 | **6.40** | 0.86 | 0.162 | **11.15** | 1.28 | 0.232 | **16.96** | 1.92 | 0.445 |
| CW-Greedy | 5.06 | 0.85 | 0.002 | 7.22 | 0.90 | 0.011 | 12.85 | 1.33 | 0.052 | 19.72 | 1.92 | 0.186 |
| CW-Rnd(5,5) | 4.86 | 0.82 | 0.016 | 6.89 | 0.84 | 0.053 | 12.35 | 1.27 | 0.217 | 19.09 | 1.85 | 0.735 |
| CW-Rnd(10,10) | 4.80 | 0.82 | 0.079 | 6.81 | 0.82 | 0.256 | 12.25 | 1.25 | 0.903 | 18.96 | 1.85 | 3.171 |
| SW-Basic | 5.42 | 0.95 | 0.001 | 7.59 | 0.93 | 0.006 | 13.61 | 1.23 | 0.096 | 21.01 | 1.51 | 1.341 |
| SW-Rnd(5) | 5.07 | 0.87 | 0.004 | 7.17 | 0.85 | 0.029 | 13.09 | 1.12 | 0.472 | 20.47 | 1.41 | 6.32 |
| SW-Rnd(10) | 5.00 | 0.87 | 0.008 | 7.08 | 0.84 | 0.062 | 12.96 | 1.12 | 0.988 | 20.33 | 1.39 | 12.443 |
| OR-Tools | **4.67** | 0.81 | 0.004 | 6.43 | 0.86 | 0.010 | 11.31 | 1.29 | 0.053 | 17.16 | 1.88 | 0.231 |
| Optimal | 4.55 | 0.78 | 0.029 | 6.10 | 0.79 | 102.8 | | — | | | — | |
| RL-SD-Greedy | 4.80 | 0.83 | 0.059 | 6.51 | 0.84 | 0.107 | 11.32 | 1.27 | 0.176 | 17.12 | 1.90 | 0.310 |
| RL-SD-BS(5) | 4.69 | 0.80 | 0.063 | 6.40 | 0.85 | 0.145 | 11.14 | 1.25 | 0.226 | 16.94 | 1.88 | 0.401 |
| RL-SD-BS(10) | 4.65 | 0.79 | 0.074 | 6.34 | 0.80 | 0.155 | 11.08 | 1.24 | 0.250 | 16.86 | 1.87 | 0.477 |

Path length = 5.72          Path length = 6.36          Path length = 10.72

Path length = 5.62          Path length = 6.06          Path length = 10.70

(a) Example 1: VRP20;       (b) Example 2: VRP20;       (c) Example 3: VRP50;
    capacity 30                  capacity 30                  capacity 40

Figure 8: Sample decoded solutions for VRP20 and VRP50 using greedy (in top row) and beam-search (bottom row) decoder. The numbers inside the nodes are the demand values.

as their demands and the depot location. The solution in the second line is the tour found by the greedy decoder. In the third and fourth line, we observe how increasing the beam width helps in improving the solution quality. Finally, we present the optimal solution in the last row. In Table 4, we illustrate an example where our method has discovered a solution by splitting the demands which is, in fact, considerably shorter than the optimal solution found by solving the mixed integer programming model.

## C.5 Attention Mechanism Visualization

In order to illustrate how the attention mechanism is working, we relocated customer node 0 to different locations and observed how it affects the selected action. Figure 9 illustrates the attention in initial decoding step for a VRP10 instance drawn in part (a). Specifically, in this experiment, we let the coordinates of node 0 equal $\{0.1 \times (i, j), \forall i, j \in \{1, \cdots, 9\}\}$. In parts (b)-(d), the small bottom left square corresponds to the case where node 0 is located at [0.1,0.1] and the others have a similar interpretation. Each small square is associated with a color ranging from black to white, representing the probability of selecting the corresponding node at the initial decoding step. In part (b), we observe that if we relocate node 0 to the bottom-left of the plane, there is a positive probability of directly going to this node; otherwise, as seen in parts (c) and (d), either node 2 or 9 will be chosen with high probability. We do not display the probabilities of the other points since there is a near-0 probability of choosing them, irrespective of the location of node 0. A video demonstration of the decoding process and attention mechanism is available online at `https://youtu.be/qGKt0bB01p0`.

|            |            |            |            |
|------------|------------|------------|------------|
| (a) VRP10 instance. | (b) Point 0. | (c) Point 2. | (d) Point 9. |

Figure 9: Illustration of attention mechanism at decoding step 0. The problem instance is illustrated in part (a) where the nodes are labeled with a sequential number; labels 0-9 are the customer nodes and 10 is the depot. We place node 0 at different locations and observe how it affects the probability distribution of choosing the first action, as illustrated in parts (b)–(d).

## C.6 Experiment on Stochastic VRP

Next, we design a simulated experiment to illustrate the performance of the framework on the *stochastic VRP* (SVRP). A major difficulty of planning in these systems is that the schedules are not defined in beforehand, and one needs to deal with various customer/demand realizations on the fly. Unlike the majority of the previous literature which only considers one stochastic element (e.g., customer locations are fixed, but the demands can change), we allow the customers and their demands to be stochastic, which makes the problem intractable for many classical algorithms. (See the review of SVRP by Ritzinger et al. [29].) We consider an instance of the SVRP in which customers with random demands arrive at the system according to a Poisson process; without loss of generality we assume the process has rate 1. Similar to previous experiments, we choose each new customer's location uniformly on the unit square and its demand to a discrete number in $\{1, \cdots, 9\}$. We fix the depot position to $[0.5, 0.5]$. A vehicle is required to satisfy as much demand as possible in a time horizon with length 100 time units. To make the system stable, we assume that each customer cancels its demand after having gone unanswered for 5 time units. The vehicle moves with speed 0.1 per time unit. Obviously, this is a continuous-time system, but we view it as a discrete-time MDP where the vehicle can make decisions at either the times of customer arrivals or after the time when the vehicle reaches a node.

The network and its hyper-parameters in this experiment are the same as in the previous experiments. One major difference is the RL training method, where we use asynchronous advantage actor-critic (A3C) [27] with one-step reward accumulation. The main reason for choosing this training method is that REINFORCE is not an efficient algorithm in dealing with the long trajectories. The details

Table 3: Solutions found for a sample VRP10 instance. We use different decoders for producing these solutions; the optimal route is also presented in the last row.

| |
|---|
| **Sample instance for VRP10**: <br> Customer locations: [[0.411, 0.559], [0.874, 0.302], [0.029, 0.127], [0.188, 0.979], [0.812, 0.330], [0.999, 0.505], [0.926, 0.705], [0.508, 0.739], [0.424, 0.201], [0.314, 0.140]] <br> Customer demands: [2, 4, 5, 9, 5, 3, 8, 2, 3, 2] <br> Depot location: [0.890, 0.252] |
| **Greedy decoder**: <br> Tour Length: 5.305 <br> Tour: $10 \to 5 \to 6 \to 4 \to 1 \to 10 \to 7 \to 3 \to 0 \to 8 \to 9 \to 10 \to 2 \to 10$ |
| **BS decoder with width 5**: <br> Beam tour lengths: [5.305, 5.379, 4.807, 5.018, 4.880] <br> Best beam: 2, Best tour length: 4.807 <br> Best tour: $10 \to 5 \to 6 \to 4 \to 1 \to 10 \to 7 \to 3 \to 0 \to 10 \to 8 \to 2 \to 9 \to 10$ |
| **BS decoder with width 10**: <br> Beam tour lengths: [5.305, 5.379, 4.807, 5.0184, 4.880, 4.800, 5.091, 4.757, 4.8034, 4.764] <br> Best beam: 7, Best tour length: 4.757 <br> Best tours: $10 \to 5 \to 6 \to 1 \to 10 \to 7 \to 3 \to 0 \to 4 \to 10 \to 8 \to 2 \to 9 \to 10$ |
| **Optimal**: <br> Optimal tour length: 4.546 <br> Optimal tour: $10 \to 1 \to 10 \to 2 \to 3 \to 8 \to 9 \to 10 \to 0 \to 4 \to 5 \to 6 \to 7 \to 10$ |

Table 4: Solutions found for a sample VRP10 instance where by splitting the demands, our method significantly improves upon the "optimal" (of which no split demand is allowed).

| |
|---|
| **Sample instance for VRP10**: <br> Customer locations: [[0.253, 0.720], [0.289, 0.725], [0.132, 0.131], [0.050, 0.609], [0.780, 0.549], [0.014, 0.920], [0.624, 0.655], [0.707, 0.311], [0.396, 0.749], [0.468, 0.579]] <br> Customer demands: [5, 6, 3, 1, 9, 8, 9, 8, 7, 7] <br> Depot location: [0.204, 0.091] |
| **Greedy decoder**: <br> Tour Length: 5.420 <br> Tour: $10 \to 7 \to 4 \to 9 \to 10 \to 6 \to 9 \to 8 \to 10 \to 1 \to 0 \to 5 \to 3 \to 10 \to 2 \to 10$ |
| **BS decoder with width 5**: <br> Beam tour lengths: [5.697, 5.731, 5.420, 5.386, 5.582] <br> Best beam: 3, Best tour length: 5.386 <br> Best tour: $10 \to 7 \to 4 \to 6 \to 10 \to 6 \to 8 \to 9 \to 10 \to 1 \to 0 \to 5 \to 3 \to 10 \to 2 \to 10$ |
| **BS decoder with width 10**: <br> Beam tour lengths: [5.697, 5.731, 5.420, 5.386, 5.362, 5.694, 5.582, 5.444, 5.333, 5.650 ] <br> Best beam: 8 , Best tour length: 5.333 <br> Best tours: $10 \to 7 \to 4 \to 9 \to 10 \to 9 \to 6 \to 8 \to 10 \to 1 \to 0 \to 5 \to 3 \to 10 \to 2 \to 10$ |
| **Optimal**: <br> Optimal tour length: 6.037 <br> Optimal tour: $10 \to 5 \to 7 \to 10 \to 9 \to 10 \to 2 \to 10 \to 8 \to 10 \to 1 \to 3 \to 4 \to 6 \to 10$ |

of the training method are described in Appendix D. The other difference is that instead of using masking, at every time step, the input to the network is a set of available locations which consists of the customers with positive demand, the depot, and the vehicle's current location; the latter decision allows the vehicle to stop at its current position, if necessary. We also add the *time-in-system* of customers as a dynamic element to the attention mechanism; it will allow the training process to learn customer abandonment behavior.

We compare our results with three other strategies: *(i) Random*, in which the next destination is randomly selected from the available nodes and it is providing a "lower bound" on the performance; *(ii) Largest-Demand*, in which the customer with maximum demand will be chosen as the next destination; and *(iii) Max-Reachable*, in which the vehicle chooses the node with the highest demand while making sure that the demand will remain valid until the vehicle reaches the node. In all strategies, we force the vehicle to route to the depot and refill when its load is zero. Even though

simple, these baselines are common in many applications. Implementing and comparing the results with more intricate SVRP baselines is an important future direction.

Table 5 summarizes the average demand satisfied, and the percentage of the total demand that this represents, under the various strategies, averaged over 100 test instances. We observe that A3C outperforms the other strategies. Even though A3C does not know any information about the problem structure, it is able to perform better than the Max-Reachable strategy, which uses customer abandonment information.

Table 5: Satisfied demand under different strategies.

| Method | Random | Largest-Demand | Max-Reachable | A3C |
|---|---|---|---|---|
| Avg. Dem. | 24.83 | 75.11 | 88.60 | 112.21 |
| % satisfied | 5.4% | 16.6% | 19.6% | 28.8% |

# D  Training Policy Gradient Methods

We utilize the REINFORCE method, similar to Bello et al. [4] for solving the TSP and VRP, and A3C [27] for the SVRP. In this Appendix, we explain the details of the algorithms.

Let us consider a family of problems, denoted by $\mathcal{M}$, and a probability distribution over them, denoted by $\Phi_{\mathcal{M}}$. During the training, the problem instances are generated according to distribution $\Phi_{\mathcal{M}}$. We also use the same distribution in the inference to produce test examples.

**REINFORCE Algorithm for VRP**    Algorithm 3 summarizes the REINFORCE algorithm. We have two neural networks with weight vectors $\theta$ and $\phi$ associated with the actor and critic, respectively. We draw $N$ sample problems from $\mathcal{M}$ and use Monte Carlo simulation to produce feasible sequences with respect to the current policy $\pi_{\theta}$. We adopt the superscript $n$ to refer to the variables of the $n$th instance. After termination of the decoding in all $N$ problems, we compute the corresponding rewards as well as the policy gradient in step 14 to update the actor network. In this step, $V(X_0^n; \phi)$ is the the reward approximation for instance problem $n$ that will be calculated from the critic network. We also update the critic network in step 15 in the direction of reducing the difference between the expected rewards with the observed ones during Monte Carlo roll-outs.

---
**Algorithm 3** REINFORCE Algorithm
---
1: initialize the actor network with random weights $\theta$ and critic network with random weights $\phi$
2: **for** $iteration = 1, 2, \cdots$ **do**
3:     reset gradients: $d\theta \leftarrow 0, d\phi \leftarrow 0$
4:     sample $N$ instances according to $\Phi_{\mathcal{M}}$
5:     **for** $n = 1, \cdots, N$ **do**
6:         initialize step counter $t \leftarrow 0$
7:         **repeat**
8:             choose $y_{t+1}^n$ according to the distribution $\pi(\cdot | Y_t^n, X_t^n; \theta)$
9:             observe new state $X_{t+1}^n$
10:            $t \leftarrow t + 1$
11:        **until** termination condition is satisfied
12:        compute reward $R^n = R(Y^n, X_0^n)$
13:    **end for**
14:    $d\theta \leftarrow \frac{1}{N} \sum_{n=1}^{N} (R^n - V(X_0^n; \phi)) \nabla_{\theta} \log P(Y^n | X_0^n; \theta)$
15:    $d\phi \leftarrow \frac{1}{N} \sum_{n=1}^{N} \nabla_{\phi} (R^n - V(X_0^n; \phi))^2$
16:    update $\theta$ using $d\theta$ and $\phi$ using $d\phi$.
17: **end for**
---

**Asynchronous Advantage Actor-Critic for SVRP**    The *Asynchronous Advantage Actor-Critic* (A3C) method proposed in [27] is a policy gradient approach that has been shown to achieve super-human performance playing Atari games. In this paper, we utilize this algorithm for training the

**Algorithm 4** Asynchronous Advantage Actor-Critic (A3C)

---

1: initialize the actor network with random weights $\theta^0$ and critic network with random weights $\phi^0$ in the master thread.
2: initialize $N$ thread-specific actor and critic networks with weights $\theta^n$ and $\phi^n$ associated with thread $n$.
3: **repeat**
4:    **for** each thread $n$ **do**
5:       sample a instance problem from $\Phi_{\mathcal{M}}$ with initial state $X_0^n$
6:       initialize step counter $t^n \leftarrow 0$
7:       **while** episode not finished **do**
8:          choose $y_{t+1}^n$ according to $\pi(\cdot|Y_t^n, X_t^n; \theta^n)$
9:          observe new state $X_{t+1}^n$;
10:         observe one-step reward $R_t^n = R(Y_t^n, X_t^n)$
11:         let $A_t^n = \left( R_t^n + V(X_{t+1}^n; \phi^n) - V(X_t^n; \phi^n) \right)$
12:         $d\theta^0 \leftarrow d\theta^0 + A_t^n \nabla_\theta \log \pi(y_{t+1}^n|Y_t^n, X_t^n; \theta^n)$
13:         $d\phi^0 \leftarrow d\phi^0 + \nabla_\phi \left( A_t^n \right)^2$
14:         $t^n \leftarrow t^n + 1$
15:       **end while**
16:    **end for**
17:    periodically update $\theta^0$ using $d\theta^0$ and $\phi^0$ using $d\phi^0$
18:    $\theta^n \leftarrow \theta^0, \phi^n \leftarrow \phi^0$
19:    reset gradients: $d\theta^0 \leftarrow 0, d\phi^0 \leftarrow 0$
20: **until** training is finished

---

policy in the SVRP. In this architecture, we have a central network with weights $\theta^0, \phi^0$ associated with the actor and critic, respectively. In addition, $N$ agents are running in parallel threads, each having their own set of local network parameters; we denote by $\theta^n, \phi^n$ the actor and critic weights of thread $n$. (We will use superscript $n$ to denote the operations running on thread $n$.) Each agent interacts with its own copy of the VRP at the same time as the other agents are interacting with theirs; at each time-step, the vehicle chooses the next point to visit and receives some reward (or cost) and then goes to the next time-step. In the SVRP that we consider in this paper, $R_t$ is the number of demands satisfied at time $t$. We note that the system is basically a continuous-time MDP, but in this algorithm, we consider it as a discrete-time MDP running on the times of system state changes $\{\tau_t : t = 0, \cdots \}$; for this reason, we normalize the reward $R_t$ with the duration from the previous time step, e.g., the reward is $R_t/(\tau_t - \tau_{t-1})$. The goal of each agent is to gather independent experiences from the other agents and send the gradient updates to the central network located in the main thread. In this approach, we periodically update the central network weights by accumulated gradients and send the updated weight to all threads. This asynchronous update procedure leads to a smooth training since the gradients are calculated from independent VRP instances.

Both actor and critic networks in this experiment are exactly the same as the ones that we employed for the classical VRP. For training the central network, we use RMSProp optimizer with learning rate $10^{-5}$.