[Reviews · NeurIPS 2018]

Reviewer 1



Many combinatorial optimization problems are only solvable exactly for small problem sizes, so various heuristics are used to find approximate solutions for larger problem sizes. Recently, there have been a number of attempts to use neural networks to learn these heuristics. This work is focused on the vehicle routing problem, a generalization of the well-known traveling salesman problem and task of significant real world interest. The solution explored in the paper is to use standard RL techniques (REINFORCE and A3C) with a slightly modified pointer net architecture. The modification is that the encoder is feedforward convolutional network rather than an RNN, meaning the network is invariant to the ordering of the input sequence. The empirical results are quite impressive, they find that this approach is able to find solutions superior to that of well-known heuristics and a baseline library of heuristics in 100\% of cases for the larger problem size (200 nodes). This work is appropriately written and cites relevant prior work. The primary weakness of this work is that it may be of more limited significance. The prior work of Bello demonstrated that RL with pointer networks were capable of outperform OR Tools on the TSP problem. This work is a generalisation of that work (with a minor architectural changes). There are two questions I have that would merit discussion in the paper. - Bello et al. use active search (further RL training on a specific problem instance) in order to iteratively search for a better solution. Was this approach tried here / why was it not used here? - How generalizable are the solutions learned for problems outside the training distribution. For example VRP100 on tasks of size 99 or 101? This seems like an important question for practical applications. Minor issues: - Although possible rewards mentioned are the negative total vehicle distance or average service time, the reward function actually used is, as far as I could find, never explicitly stated. - The policy is denoted as both $\pi$ (standard RL notation) and, to emphasize the sequential structure $P(Y|X_0)$. This change seems distracting. -- The author's response demonstrated some level of generalisation in problem size and explained why they are not using active search (although it would be nice to see this as a baseline). Given the generalisation results I've increased my rating.

Reviewer 2



This paper takes part in this rather recent (renewed would be more exact) attempt to solve combinatorial optimization (CO) problems with (deep) reinforcement learning. Here, the authors build on previous works to provide a novel, quite simple, approach to solve the vehicle routing problem (VRP). VRP is a quiet complex CO problem, much more complex than TSP and Knapsack problems. There are many variants of the VRP. Here, the authors deal with the one in which a single vehicle of limited capacity has to deliver items to a set of geographically distributed customers, each with finite demand. The general idea is to train a reinforcement learner to solve any VRP instance drawn from a certain distribution. Overall I find this paper well written and rather clear. However, I don't completely understand whether training depends on the number of delivery locations or not. The experiments are quite convincing. They compare their approach with algorithms like Clarke-Wright that can be considered as a baseline, sweep heuristics (this one deserves explanation or a bibliographical reference), and Google's OR-tools. The proposed approach obtains the best performance along with Google OR-Tools on small VRP instances (10 and 20 delivery locations). For larger instances, the proposed approach is often the best, though OR-Tools is a strong competitor. Regarding computation times, I find figure 4 quite unclear but I feel like OR-Tools is also faster than the proposed approach. However, it seems that its computation time scales faster than for the proposed RL one. But I really don't find the figure clear nor informative: with simple numbers, what are the wall clock times for both approaches. Minor remark: regarding figure 4, I doubt the figures mentioned on the ordinate axis are logs (though the scale is logarithmic). The code is not provided. Reproducing the results is quite unlikely to succeed without further details. Overall, I find this paper an interesting and significant contribution to the state of the art. I am not sure that it contributed to the state of the art on the VRP; this is a point that would deserve to be clarified: are we making progress towards solving VRP with RL for real applications, or is there yet a significant gap between the state-of-the-art algorithms to solve VRP and a learning approach?

Reviewer 3



Updated review: ---------------------- I am happy with the authors response to my two main concerns. I am raising my score to an accept conditional on the additional results being added to the paper. Original review: --------------------- The reviewed paper presents an application of model-free reinforcement learning to the problem of optimal vehicle routing. The considered application is interesting and potentially of high-impact. It is computationally difficult to solve and finding optimal solutions for large instances is - "in the wild" - currently done with the use of heuristics. The presented appproach marks a significant improvement over said heuristics while being "cheap" to compute. The paper is well written and easy to follow, the related work background is reasonable, yet it fails to point out connections to graph-networks. The description of problem setting is thorough and easily understandable, making this paper a good reference for future attempts to address similar problems with ML techniques. The contribution of the paper is purely empirical - which is not necessarily a bad thing. It builds on existing theory and proposes problem specific adjustments. The empirical evaluation is thorough and meaningful, giving useful comparisons to externally implemented SOTA algorithms as well as to simple heuristics. Arguably such a paper might be better suited for presentation in an operations research conference or journal and not perfectly suited for NIPS. I however refrain from such judgement as I think the paper is interesting enough (modulo my main concerns below) and would suggest that the area chair has a second look. Overall the paper could well be accepted to NIPS if the two main concerns that I outline below would prove to be unsubstantiated / could be addressed. For now I mark it with a weak reject but I would increase my score to a weak accept/accept if the concerns can be addressed (which would merely require running additional experiments with Pointer networks and adding a short paragraph in the related work). Main concerns with the paper: 1) Section 3.1 makes it sound as if pointer networks are simply not applicable to the problem. Which is not the case as far as I understand. Sure it will be more expensive to re-evaluate the pointer network once an input has changed but it is technically possible. I fail to see why this cannot be implemented (and in fact just changing the inputs every time step e.g. in tensorflow should automatically generate the required computational graph). In fact you mention in l. 189 that the problem with applying a simple Pointer Network is mainly a computational one. If this is the case then you should provide a comparison to pointer networks at least on small scale instances and show when and how they break! 2) Your proposed solution in 3.2, to me, reads simply as applying a special case of a graph network (see e.g. [1] and maybe [2-3] for introductions) how is this different than a graph network with a pointer network output ? If it is not then you should discuss/relate to the graph-network line of work in the related work/background section. Experiments: - What is the variance of the results wrt. training seeds ? I.e. the plot in Figure 3 seems to show variance accross problem instances for one trained policy, but how does the performance vary wrt. the training itself ? - One appealing property of the method should be that it can handle variable problem sizes yet no attempt is made to quantify this property. Why do you not try to train on VRP10 to VRP100 and see how well a joint model does ? Would such variability in training sequence size not anyway be required for any practical application ? If so it would also be important to know how the policy compares in these instances to e.g. OR-Tools. Unfortunately does not generalize over problem instances, e.g. more nodes etc. Specific comments and minor typos: l. 29.: "milestone on the path toward approaching" -> I guess you mean milestone on the path towards solving ? l. 34-36: "Obviously, this approach is not practical in terms of either solution quality or runtime since there should be many trajectories sampled from one MDP to be able to produce a near-optimal solution." -> It is not clear to me what you mean here, I guess you mean that solving an MDP "on-the-fly" every time demands change is too expensive ? What do you mean by many trajectories are needed ? It is not clear that you could not use a model for this to me and what the exact difference is you are trying to point out here. Revise that sentence. l. 52: "which is actually simpler than the Pointer Network approach" -> Remove actually. This sounds as if there was a counter-point to said claim before. Expressing your surprise here is not necessary. l. 54: "Our model" -> it is confusing to call this a model here and before as you are *not* trying to learn a model of the MDP but just use a different parameterization for the policy. Maybe re-write to "policy model" or "actor-critic model" or something like that ? l. 59: appealing to whom ? I guess you mean it is appealing to practitioners ? l. 64: "inputs change in any way" -> clearly not in any arbitrary way :) try to be a bit more precise, e.g. do you want to say it is robust to noise or to changing rewards of the VRP or ... l. 72: "is ones" -> "is one" l. 78: "and *at* no extra cost" l. 80: reads colloquial better: "[...] we briefly review required notation and relations to existing work" ? l. 84: "which is almost the same": either substantiate this claim or simply write: "which shared by many of these models" l. 91: "wisely" is inappropriate here -> chang eto "more directly" or "more explicitly" l. 98: "for policy representation" -> "for the policy parameterization" Section 3: as written above the use of the term Model is a bit loose in the paper. Sometimes it referes to the policy, somethimes to the problem formulation itself (the MDP) and sometimes to the combination of both, try to be more specific. l. 226: remove braces this is a full sentence. [1] Scarselli, F., Gori, M., Tsoi, A. C., Hagenbuchner, M., and Monfardini, G. Computational capabilities of graph neural networks [2] Li, Y., Tarlow, D., Brockschmidt, M., and Zemel, R. Gated graph sequence neural networks. [3] Battaglia, P., Pascanu, R., Lai, M., Rezende, D. J., et al. Interaction networks for learning about objects, relations and physics